# A High-Resolution Joint Angle-Doppler Estimation Sub-Nyquist Radar Approach Based on Matrix Completion

**Quanhui Wang** [1] **and Ying Sun** [2,*]

[1]    School of Information Engineering, Lingnan Normal University, Zhanjiang 524000, China; wangquanhui@lingnan.edu.cn
[2]    Huawei Technologies Co., Ltd., Shenzhen 518000, China
*    Correspondence: sun-ying1984@hotmail.com

**Abstract:**   In order to reduce power consumption and save storage capacity, we propose a high-resolution sub-Nyquist radar approach based on matrix completion (MC), termed as single-channel sub-Nyquist-MC radars. While providing the high-resolution joint angle-Doppler estimation, this proposed radar approach minimizes the number of samples in all three dimensions, that is, the range dimension, the pulse dimension (also named temporal dimension), and the spatial dimension. In range dimension, we use a single-channel analog-to-information converter (AIC) to reduce the number of range samples to one; in both spatial and temporal dimensions, we employ a bank of random switch units to regulate the AICs, which greatly reduce the number of spatial-temporal samples. According to the proposed sampling scheme, the samples in digital processing center forwarded by M receive nodes and N pulses are only a subset of the full matrix of size M times N. Under certain conditions and with the knowledge of the sampling scheme, the full matrix can be perfectly recovered by using MC techniques. Based on the recovered full matrix, this paper addresses the problem of the high-resolution joint angle-Doppler estimation by employing compressed sensing (CS) techniques. The properties and performance of the proposed approach are demonstrated via simulations.

**Keywords:** compressed sensing radar; sub-Nyquist sampling; matrix completion; array signal processing; recovery algorithm

---

## 1. Introduction

In a typical move target indication (MTI) radar scenario [1], matched filtering (MF) is done separately on the returns from each pulse of each receive node, after which the signals are sampled by the analog-to-digital converter (ADC) and sent to a digital processing center, as illustrated in Figure 1. The digital processing center performs all subsequent radar signal and data processing, such as target detection and parameter estimation [2]. For each pulse repetition interval (PRI), $L$ range samples are collected to cover the range interval. With $M$ receive nodes and $N$ pulses, the received data for one coherent processing interval (CPI) comprises $L \times M \times N$ complex samples. However, growing demands for target distinction capability imply significant growth in both the number of channels and the signal bandwidth in modern radar systems [3–7]. Under the confinement of classic bandpass sampling theorem [8], sampling at the Nyquist rate would result in considerably large snapshots $L$. Accordingly, the number of overall samples $L \times M \times N$ would be tremendous, implying an increase in potential power consumption and also the requirement of huge storage capacity for subsequent processing. Therefore, real-time processing would be quite difficult in MTI radar systems.

For these reasons, we expect to communicate the least samples as possible to the digital processing center while preserving the ability to perfectly reconstruct the signal. Inspired by this motivation, we propose a high-resolution sub-Nyquist radar approach based on matrix completion (MC), termed as single-channel sub-Nyquist-MC radars. This proposed radar approach minimizes the number of samples in all three dimensions: the range dimension, the pulse dimension, and the spatial dimension. In range dimension, we use a single-channel analog-to-information converter (AIC) to reduce the number of samples from $L$ to 1. AIC is a technique for sampling analog signals directly at a rate lower than Nyquist sampling rate [9–11]; in both pulse dimension and spatial dimension, we employ a bank of random switch units to regulate the AICs, which reduces the number of samples from $M \times N$ to $m$ ($m \ll M \times N$). Consequently, the number of total samples is reduced from $L \times M \times N$ to $m$ by adopting the proposed radar approach.

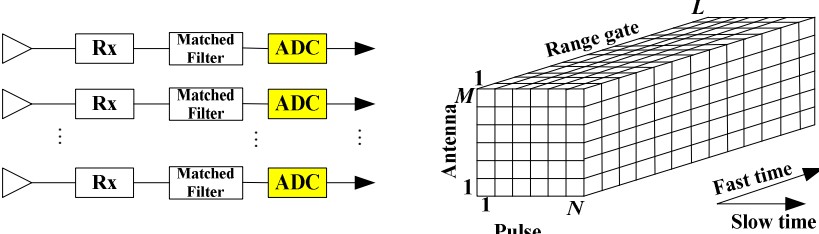

**Figure 1.** Processing for receive nodes (left) and the coherent processing interval (CPI) data cube (right) in conventional move target indication (MTI) radar systems.

In order to avoid information loss, the sampling rate cannot be simply reduced. Therefore, we adopt sub-Nyquist radar techniques in the range dimension. Sub-Nyquist radar allows sampling at rates much lower than Nyquist while being able to exactly recover the signal, which has been demonstrated by real-time analog experiments in hardware [12,13]. For example, Ref. [14] proposed a sub-Nyquist sampling approach called the direct multichannel sampling scheme, in which an analog prefiltering operation is performed and then sampled in order to extract the required information for recovery. The random demodulation (RD) [11] and the modulation bandwidth converter (MWC) [15] are the mainstream implementations of AIC. The RD scheme multiplies the signal and the pseudo-random chip sequence by mixing and low-pass filtering. Sampling is then performed by a low-speed analog-to-digital converter that is lower than the Nyquist rate. Reference [15] proposed an MWC scheme based on compressed sensing. Signals are captured with an analog front end that consists of a bank of multipliers and low-pass filters whose cutoff is much lower than the Nyquist rate. To minimize the samples while avoiding information loss, we use a single-channel AIC to perform the sampling in range dimension, which is shown in Figure 2. In each receive channel, both the matched filter and high-rate ADC in MTI radars are replaced by an AIC, before which a random switch unit is used to turn on and turn off the AIC. This scheme implies that only one sample can be obtained at each receive node during one pulse when the parallel random switch unit is turned on. According to the proposed sampling scheme, the samples in the digital processing center forwarded by all receive nodes and all pulses are only a subset of the entries of a $M \times N$ matrix. If and only if all switch units are turned on, the samples can be arranged into a full matrix of size $M \times N$. When the number of target $K$ is much smaller than the number of receive nodes $M$ and the number of pulses $N$, the full matrix is of low rank. This means that, under certain conditions and with the knowledge of the sampling scheme, the full matrix can be exactly recovered by using matrix completion techniques based on the observations of a small subset of the full matrix. There are several recent papers on matrix completion problem [16–23]. For example, Candes and Recht proved that most low-rank matrices could be recovered exactly from most sets of sampled entries even though these sets have surprisingly small cardinality, and more importantly, they proved that this could be done by solving a simple convex optimization problem [16]. Cai et al. proposed the singular value thresholding (SVT) algorithm, which can also directly recover

an unknown (approximately) low-rank matrix from very limited information and have minimum storage requirement since only principle factors are needed to keep in memory [23]. MC techniques furthermore have been applied as means of reducing the volume of data required in Multiple-Input Multiple-Output (MIMO) radars for target detection and estimation [24–26], while the theoretical results and performance bounds for Scheme I of MIMO-MC radar are derived in [27].

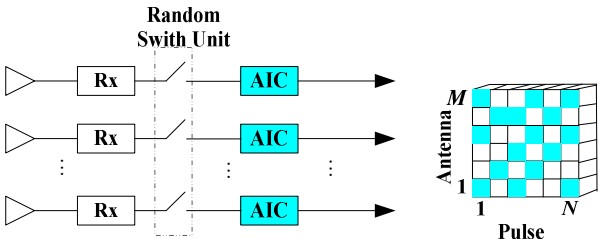

**Figure 2.** The proposed single-channel sub-Nyquist radar approach (**left**) and a subset of the full matrix (**right**).

Based on the recovered full matrix, this paper tackles the high-resolution joint angle-Doppler estimation problem. This estimation comes down to an underdetermined equation solving the problem. In order to apply the compressed sensing (CS) techniques, we discretize the angle-Doppler plane into a discrete grid. This gridding approach is first proposed for CS radar in which the time-frequency plane is discretized into an $N \times N$ grid [3,5]. After that, this idea is extended to the MIMO radar [28] when the signals are sparse in the range-Doppler-angle space. Since the target signal is sparse in the angle-Doppler domain, a variety of CS [29,30] techniques can be employed for recovery, for instance, orthogonal matching pursuit (OMP) [31] and iterative hard thresholding (IHT) [32]. Finally, we are able to obtain estimates of the angles and Doppler frequencies of targets by using a compressed sensing recovery algorithm. The properties and performance of the proposed approach are demonstrated via simulations. Compared to conventional MTI radar, the proposed radar approach has the advantage in terms of the most significant reduction in the number of samples needed for accurate joint angle-Doppler estimation. In a scenario with $K$ point targets in the far field, when the number of samples is reduced from $L \times M \times N$ required by conventional MTI radar to $m$, where $m$ is the number of samples, $m \approx 4df$ and $df = K(M + N - K)$ [23], the proposed single-channel sub-Nyquist-MC radar is still able to achieve high-resolution angle-Doppler estimation.

A. Notation

Lower-case and upper-case letters in bold denote vectors and matrices, respectively. Superscripts $(\cdot)^T$ and $(\cdot)^H$ denote transpose and Hermitian transpose, respectively. $\|\mathbf{X}\|_*$ is the nuclear norm, that is, the sum of the singular values. $\|\mathbf{X}\|_F$ is the Frobenius norm. $\|\mathbf{x}\|_0$ means the $l_0$-norm, that is, a total number of nonzero elements in a vector. $\|\mathbf{x}\|_1$ and $\|\mathbf{x}\|_2$ denote the $l_1$-norm and $l_2$-norm, respectively.

## 2. Methods

In this section, we first briefly introduce the required background in matrix completion and then review the fundamentals of a typical sampling approach in MTI radar systems.

Let us consider conventional MTI radar with a uniform linear array (ULA) of $M$ elements. The radar transmits a coherent burst of $N$ pulses at a constant pulse repetition frequency (PRF) $f_r = 1/T_r$, where $T_r$ is the PRI. Then, the length of a CPI is equal to $NT_r$. On the receiver, each element has its own matched filter, A/D converter, that is, matched filtering is done separately on the returns from each pulse of each element, after which the signals are sampled by the A/D converter and sent to a digital processing center. This digital processing center performs all subsequent radar signal and data processing, such as target detection and parameter estimation. $L$ range samples are collected from each pulse and each element. Hence, the received data for one CPI comprises $L \times M \times N$ complex baseband samples from $N$ pulses and $M$ receiver elements. The three-dimensional data set is often visualized

as the $L \times M \times N$ cube of complex samples shown in Figure 1. The directions along the columns and rows are referred to as a spatial dimension and pulse dimension (also called slow-time dimension), respectively. The third dimension is the range dimension, also called fast-time dimension. The data for a single range gate can be written as a the $l$th $M \times N \times 1$ vector $\mathbf{x}_l$, termed as space-time snapshot

$$\mathbf{x}_l = [x_{11l}, x_{12l}, \cdots x_{MNl}]^T \tag{1}$$

where $x_{mnl}$ denotes the complex samples from the $m$th element, $n$th pulse, and $l$th range gate.

Based on this data cube, high-resolution estimation methods can be applied to estimate targets' angles and Doppler frequencies. However, these high-resolution estimation methods require large numbers of training snapshots to maintain good performance. In order to reduce power consumption and save storage capacity, we propose a high-resolution sub-Nyquist radar approach based on matrix completion, termed as single-channel sub-Nyquist-MC radar. This proposed radar approach reduces the number of samples as much as possible in all three dimensions while providing the high-resolution joint angle-Doppler estimation.

## 3. The Proposed Single-Channel Sub-Nyquist-MC Radar Approach

Suppose that a random single-channel AIC is used to replace the matched filter and high-rate ADC in each receive channel, as shown in Figure 2, in which a random switch unit is used to turn on and off each AIC. The structure of the single-channel AIC is illustrated in Figure 3, in which the continuously received waveform over each PRI is sampled at the sub-Nyquist rate using only one direct channel sampling scheme. For each element, the analog input is mixed with the harmonic signal $e^{-j2\pi k_0 t/T_r}$ integrated over the PRI duration, and then sampled, where $k_0$ is an arbitrary positive integer less than $L$. This scheme implies that only one sample can be obtained at each receive element during one pulse when the random switch unit is turned on. This direct sampling scheme is straightforward, which can be implemented by only using oscillators, mixers, and integrators.

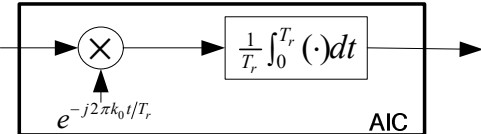

**Figure 3.** Single-channel direct sampling of the Fourier series coefficient for each receive element during one pulse.

### 3.1. ULA Case

Consider the simplest ULA case. If and only if all switch units are turned on, the samples can be arranged into a full matrix $\mathbf{X}$ of size $M \times N$. Let us consider a scenario with $K$ point targets in the far field. The $k$th target is described by its angle $\theta_k$ and Doppler frequency $f_{dk}$, $k = 1, \cdots, K$. Then, the full matrix $\mathbf{X}$ can be expressed as

$$\mathbf{X} = \mathbf{Z} + \mathbf{W} = \mathbf{A}\mathbf{D}\mathbf{B}^T + \mathbf{W} \tag{2}$$

where the matrix $\mathbf{Z}$ and $\mathbf{W}$ denote the signal and noise component, respectively; $\mathbf{D} = diag(\mathbf{d})$ with $\mathbf{d} = [\alpha_1, \cdots, \alpha_K]^T$ denotes the complex amplitudes of targets; $\mathbf{A}$ is the $M \times K$ receive steering matrix, defined as $\mathbf{A} = [\mathbf{a}(\theta_1), \mathbf{a}(\theta_2), \cdots, \mathbf{a}(\theta_K)]$; $\mathbf{B}$ is the $N \times K$ Doppler steering matrix, defined as $\mathbf{B} = [\mathbf{b}(f_{d1}), \mathbf{b}(f_{d2}), \cdots, \mathbf{b}(f_{dK})]$, where

$$\begin{aligned} \mathbf{a}(\theta_k) &= [1, e^{j\frac{2\pi}{\lambda} d \sin(\theta_k)}, \cdots, e^{j\frac{2\pi}{\lambda}(M-1)d \sin(\theta_k)}]^T \\ \mathbf{b}(f_{dk}) &= [1, e^{j2\pi f_{dk} T_r}, \cdots, e^{j2\pi f_{dk}(N-1)T_r}]^T \end{aligned} \tag{3}$$

are the spatial steering vector and temporal steering vector, respectively, where $\lambda$ and $d$ are the wavelength and the interelement spacing, respectively. The problem formulation given in (7) is similar

to that of MIMO-MC radar [24–27], where Ref. [27] provided the detailed analysis regarding the recoverability of the data matrix in collocated MIMO radar system.

According to the proposed sampling scheme illustrated in Figure 2, the samples in the digital processing center forwarded by all receive nodes and pulses are only a subset of the entries of the full matrix **X**. In the digital processing center, therefore, the observational vector **c** can be related to the full matrix **X** as the following equation

$$\mathbf{c} = \mathcal{P}_\Omega(\mathbf{X}) = \mathcal{P}_\Omega(\mathbf{Z}) + \mathcal{P}_\Omega(\mathbf{W}). \tag{4}$$

where $\mathcal{P}_\Omega(\cdot)$ is an entrywise sampling operator.

To recover the signal component **Z** in (2) with matrix completion, we need to demonstrate that the matrix **Z** indeed obeys **A0)** and **A1)**. Therefore, we show that the maximum coherence of the spaces spanned by the left and right singular vector of **Z** is bounded by the parameter $\mu_0$ (see Appendices A and B).

Consequently, the matrix recovery is done by solving the following nuclear norm optimization problem with quadratic constraint

$$\begin{aligned} &\min \|\mathbf{E}\|_* \\ &\text{s.t. } \|\mathbf{c} - P_\Omega(\mathbf{E})\| \le \delta. \end{aligned} \tag{5}$$

where $\|\mathbf{E}\|_*$ is the nuclear norm, which is the sum of the singular values.

This optimization problem can be solved by the singular value thresholding (SVT) algorithm, which is a rather powerful computational tool, especially for large scale matrix completion. The recovered data matrix $\hat{\mathbf{Z}}$ is the optimal solution $\mathbf{Z}_{opt}$ of the problem.

From the Appendix A. we know that the smaller the $\mu_0$, the fewer samples would be required to recover **Z**. Since $\xi_{u(v)} \in (0, \frac{1}{2}]$ by assumption, both the constants $\beta_{\xi_u}$ and $\beta_{\xi_v}$, as defined in (A26) and (A19), respectively, would be always finite. At this point, for sufficiently large but finite $M$ and $N$, the coherence $\mu(U)$ and the coherence $\mu(V)$ are given by

$$\mu(U) \approx \mu(V) \approx 1 \tag{6}$$

Consequently, we have $\mu_0 \ge \max(\mu(U), \mu(V)) \approx 1$.

Before we proceed with a discussion of bounds regarding the number of observations, let us state the following reconstruction theorem and lemma.

**Theorem 2 [12]:** *Let $M \in \mathbf{C}^{N_1 \times N_2}$ be a matrix of rank r obeying the strong incoherence property with parameter $\mu$ and set $N \triangleq \max\{N_1, N_2\}$. Suppose we observe m entries of $M$ with locations sampled uniformly at random. Then, there exist positive numerical constants $C_1$ and $C_2$ such that if*

$$\begin{aligned} m &\ge C_1 \mu^4 N r^2 \log^2 N \quad or \\ m &\ge C_2 \mu^2 N r \log^6 N \end{aligned} \tag{7}$$

*the minimizer to the Equation (A2) is unique and equal to $M$ with probability at least $1 - N^{-3}$.*

**Lemma 1 [24]:** *If a matrix $M$ of rank r is incoherent with parameter $\mu_0$ and $\mu_1$, it is strongly incoherent with parameter $\mu \le \mu_0 \sqrt{r}$.*

Hence, using Lemma 1, for a fixed number of targets $K$, **Z** is strongly incoherent with parameter

$$\mu \le \mu_0 \sqrt{K} \approx \sqrt{K} \tag{8}$$

Set $G \triangleq \max\{M, N\}$. Combining Theorem 1, therefore, there exist positive numerical constants $C_1$ and $C_2$ such that if

$$m \geq C_1 K^4 G \log^2 G \quad \text{or} \\ m \geq C_2 K^2 G \log^6 G \tag{9}$$

the minimizer to the Equation (A2) is unique and equal to $\mathbf{Z}$ with probability at least $1 - G^{-3}$.

That is, for a sufficiently large number of elements and a large number of pulses, and for a fixed and relatively small number of targets, matrix completion is exact if the number of observation is at least of the order of the number of observation $G \text{polylog}(G)$. But for a finite number of elements and number of pulses, the simulation results presented in Section 5 show that matrix completion is exact when the number of observation approximately equals to $4df$, where $df = K(M + N - K)$.

## 3.2. Arbitrary 2-D Array Case

We extended the analysis of coherence of $\mathbf{Z}$ for the arbitrary 2-dimensional array case. Since the pulse dimension is not changed, we only focus on the coherence $\mu(U)$ of $\mathbf{Z}$. Consider an arbitrary array equipped with $M$ antennas. Assume that the set of targets $\{\theta_k\}_{k \in \mathbf{N}_{K-1}}$ consists of almost surely distinct members and that

$$(\theta_i, \theta_j) \in \subseteq \mathbf{R}^2 - \left\{ (x, y) \in \mathbf{R}^2 \big| x \neq y \right\} \tag{10}$$

$\forall (i, j) \in \mathbf{N}_{K-1} \times \mathbf{N}_{K-1}$ with $i \neq j$, where constitutes a nominal point set for all admissible angle pair combinations. At this point, the $M \times K$ receive steering matrix $\mathbf{A}$ defined as $\mathbf{A} = [\mathbf{a}(\theta_1), \mathbf{a}(\theta_2), \cdots, \mathbf{a}(\theta_K)]$, where

$$\mathbf{a}(\theta_k) = \left[ 1, e^{j2\pi \mathbf{r}^T(1)\mathbf{\Gamma}(\theta_k)}, \cdots, e^{j2\pi \mathbf{r}^T(M-1)\mathbf{\Gamma}(\theta_k)} \right]^T \tag{11}$$

where

$$\mathbf{r}(m) \triangleq \frac{1}{\lambda} [x_m, y_m]^T \in \mathbf{R}^{2\times 1}, \quad m \in \mathbf{N}_{M-1} \tag{12}$$

$$\mathbf{\Gamma}(\theta) \triangleq [\cos(\theta), \sin(\theta)]^T \in \mathbf{R}^{2\times 1}, \tag{13}$$

with the collection of vectors $\left\{ [x_m, y_m]^T \right\}_{m \in \mathbf{N}_{M-1}}$ denoting the 2-dimensional antenna coordinates of the arbitrary array.

Similarly, we still have the almost surely rank-$K$ matrix

$$\mathbf{Z} = \mathbf{A}\mathbf{D}\mathbf{B}^T \in \mathbf{C}^{M\times N} \tag{14}$$

In order to derive the strictly positive lower bound for $\lambda_{\min}(\mathbf{A}^H\mathbf{A})$, we have,

$$\mathbf{A}^H\mathbf{A} \triangleq \begin{bmatrix} M & \delta_{1,0} & \cdots & \delta_{K-1,0} \\ \delta_{1,0}^* & M & \cdots & \delta_{K-1,1} \\ \vdots & \vdots & \ddots & \vdots \\ \delta_{K-1,0}^* & \delta_{K-1,1}^* & \cdots & M \end{bmatrix} \tag{15}$$

where

$$\delta_{i,j} \triangleq \sum_{m=0}^{M-1} e^{j2\pi r^T(m)(\mathbf{\Gamma}(\theta_i) - \mathbf{\Gamma}(\theta_j))}, \quad \forall (i, j) \in \mathbf{N}_{K-1} \times \mathbf{N}_{K-1} \tag{16}$$

with $M \equiv \delta_{i,i}, \forall i \in \mathbf{N}_{K-1}$.

Define $\mathbf{M} \triangleq \mathbf{A}^H\mathbf{A} \in \mathbf{C}^{K\times K}$. The trace of $\mathbf{M}$ is simply $MK$. Hence, we have

$$\tau = \frac{MK}{K} = M \tag{17}$$

We also need the trace of $\mathbf{M}^2$. Since $\mathbf{M}$ is a Hermitian matrix, it is true that

$$
\begin{aligned}
\mathrm{tr}\!\left(\mathbf{M}^2\right) &= \sum_{k_1=0}^{K-1}\sum_{k_2=0}^{K-1}\left|\delta_{k_1,k_2}\right|^2 \\
&\equiv \sum_{k_1=0}^{K-1}\left\{ M^2 + \sum_{\substack{k_2=0 \\ k_1\neq k_2}}^{K-1}\left|\sum_{m=0}^{M-1}e^{j2\pi r^T(m)(\Gamma(\theta_{k_1})-\Gamma(\theta_{k_2}))}\right|^2\right\}
\end{aligned}
\tag{18}
$$

Then, we can define a bivariate function $\varphi_u :\to \mathbb{C}$, given by

$$
\varphi_u(x,y) \triangleq \sum_{m=0}^{M-1}e^{j2\pi r^T(m)(\Gamma(x)-\Gamma(y))}
\tag{19}
$$

Whose norm can be bounded as

$$
\left|\varphi_u(x,y)\right|^2 \leq \sup_{(x,y)\in A}\left|\varphi_u(x,y\big|\mathfrak{I})\right|^2 \in [0,M^2)
\tag{20}
$$

Then, we can bound $\mathrm{tr}\!\left(\mathbf{M}^2\right)$ as

$$
\begin{aligned}
\mathrm{tr}\!\left(\mathbf{M}^2\right) &\leq \sum_{k_1=0}^{K-1}\left\{ M^2 + (K-1)\sup_{(x,y)\in}\left|\varphi_u(x,y)\right|^2\right\} \\
&\equiv KM^2 + K(K-1)\beta_a
\end{aligned}
\tag{21}
$$

where

$$
\beta_a \triangleq \sup_{(x,y)\in}\left|\varphi_u(x,y)\right|^2
\tag{22}
$$

At this point, in the arbitrary array case, the associated matrix $\mathbf{Z}$ obeys the assumptions **A0)** and **A1)** with

$$
\mu_0 \triangleq \max\left\{\frac{M}{M-(K-1)\sqrt{\beta_a(M)}},\frac{N}{N-(K-1)\sqrt{\beta_{\xi_v}(N)}}\right\}
\tag{23}
$$

with probability 1. That is, the proposed approach still works if the radar system is equipped with an arbitrary 2-D array, such as an identical uniform circular array (UCA).

## 4. Joint Angle-Doppler Estimation with Recovered Matrix

Based on the recovered matrix $\hat{\mathbf{Z}}$, this section addresses the high-resolution joint angle-Doppler estimation problem. Vectorizing the recovered matrix $\hat{\mathbf{Z}}$ by stacking each succeeding column one beneath the other yields a single space-time snapshot $\mathbf{z}$, which can be expressed as

$$
\mathbf{z} = \sum_{k=1}^{K}\alpha_k u_k + \mathbf{w}
\tag{24}
$$

where $\mathbf{w}$ is noise vector, $\alpha_k$ is the $k$th target's complex parameter accounting for both channel effects and target radar cross section (RCS), and the steering vector $\mathbf{u}_k$ can be written as

$$
\mathbf{u}(\theta_k,f_{dk}) = \mathbf{a}(\theta_k)\otimes\mathbf{b}(f_{dk}).
\tag{25}
$$

In order to apply the CS techniques, we set up a discrete space-time (angle-Doppler frequency) grid $\left\{(\theta_i,f_{dj})\right\}$, $1\leq i\leq N_\theta$, $1\leq j\leq N_D$, where $N_\theta$ and $N_D$ are the resolution of angle and Doppler

frequency, respectively. This gridding approach is employed for CS based (MIMO) radar to reconstruct the target scene [2–5,28]. Using vector $\mathbf{u}(\theta, f_d)$ for all grid points $\{(\theta_i, f_{dj})\}$, we construct a complete response matrix $\mathbf{H}$ whose columns are $\mathbf{u}(\theta_i, f_{dj})$ for $1 \le i \le N_\theta$ and $1 \le j \le N_D$. Based on that, the signal component can be represented by some points in the discrete angle-Doppler plane, that is, the vector $\mathbf{z}$ can be rewritten as

$$\mathbf{z} = \mathbf{H}\boldsymbol{\beta} + \mathbf{w} \tag{26}$$

where $\mathbf{H} \in \mathbb{C}^{MN \times N_\theta N_D}$ is the dictionary matrix, which represents all possible angle and Doppler frequency of the interest targets, and $\boldsymbol{\beta} = (\beta_0, \cdots, \beta_{N_\theta N_{f_d}-1})^T$, $\boldsymbol{\beta} \in \mathbb{C}^{N_\theta N_D \times 1}$ is the coefficient vector. Because the target scene $\boldsymbol{\beta}$ is known to be sparse (i.e., $\|\boldsymbol{\beta}\|_0 \ll MN$), the compressed sensing technique is a powerful tool to recover $\boldsymbol{\beta}$. Compressed sensing reconstruction methods include using greedy algorithms, such as OMP, which is used to solve the following convex problem:

$$\hat{\boldsymbol{\beta}} = \operatorname{argmin}\left\{\|\boldsymbol{\beta}'\|_1 : \|\mathbf{z} - \mathbf{H}\boldsymbol{\beta}'\|_2^2 \le \varepsilon\right\} \tag{27}$$

## 5. Numerical Results

In this section, we demonstrate the performance of the proposed approaches in terms of matrix recovery error and the joint angle-Doppler estimation based on the recovered matrix. We use ULA for receivers. The carrier frequency is set to $f = 1 \times 10^9$ Hz, which is a typical radar frequency. The noise introduced is white Gaussian with zero mean and variance $\sigma^2$. Here, we use the SVT algorithm to recover the data matrix. This is because both the storage space and computation cost SVT algorithm are very low at its every iteration and it is suitable for a large size problem with a low-rank solution. To ensure the SVT algorithm converges, we set the thresholding parameter to $\tau = 5\xi$ in this simulation, where $\xi$ is the dimension of the low-rank matrix that needs to be recovered.

### 5.1. Matrix Recovery Error under Noisy Observations

We consider a scenario with two targets. The first target's angle $\theta_1$ and normalized Doppler frequency $f_{d1}$ are generated at random in $[-90°, 90°]$ and $[-0.5, 0.5]$, respectively, and the second target's angle and normalized Doppler frequency are taken as $\theta_2 = \theta_1 + \Delta\theta$ and $f_{d2} = f_{d1} + \Delta f_d$, respectively. The target reflection coefficients are set as complex randomly. The Signal to Noise Ratio (SNR) at each receive antenna is 15 dB. In the following, we compute the matrix recovery error as a function of the number of samples $m$ per degrees of freedom df, that is, m/df, a quantity also used in [23]. A matrix of size $M \times N$ with rank $K$ has $K(M + N - K)$ degrees of freedom [23]. Let $\phi_{\hat{\mathbf{Z}}}$ denote the relative matrix recovery error, defined as:

$$\phi_{\hat{\mathbf{Z}}} = \|\hat{\mathbf{Z}} - \mathbf{Z}\|_F / \|\mathbf{Z}\|_F. \tag{28}$$

Both Figures 4 and 5 show the relative matrix recovery error $\phi_{\hat{\mathbf{Z}}}$ as a function of the number of samples per degree of freedom for the scenario described above, where Figure 4 describes the case of different angle separation between two targets while having the same Doppler frequency, and Figure 5 provides the case of different Doppler separations between two targets while having the same angle. The number of receiver antennas and the number of pulses are, respectively, set as $M = 64$ and $N = 64$. One can observe that $\phi_{\hat{\mathbf{Z}}}$ converges very fast. It converges in about when $m/df \approx 4$ in this simulation. It can also be seen from Figures 4 and 5 that when the two targets have the same angle and the same Doppler frequency, the relative recovery error is the smallest for both the cases because the coherence parameters of data matrixes are optimum, that is, $\mu_0 = 1$. With the increase in the angle separation in Figure 4 or the Doppler frequency separation in Figure 5 between two targets, the relative recovery error increases for the case of $2 \le m/df \le 4$. Finally, from both Figures 4 and 5, it can be seen that in the noisy cases, when $m/df \ge 4$, which corresponds to the matrix occupancy ratio more than $4df/(MN) \approx 0.25$, the relative recovery errors of the matrices decrease to the reciprocal of SNR.

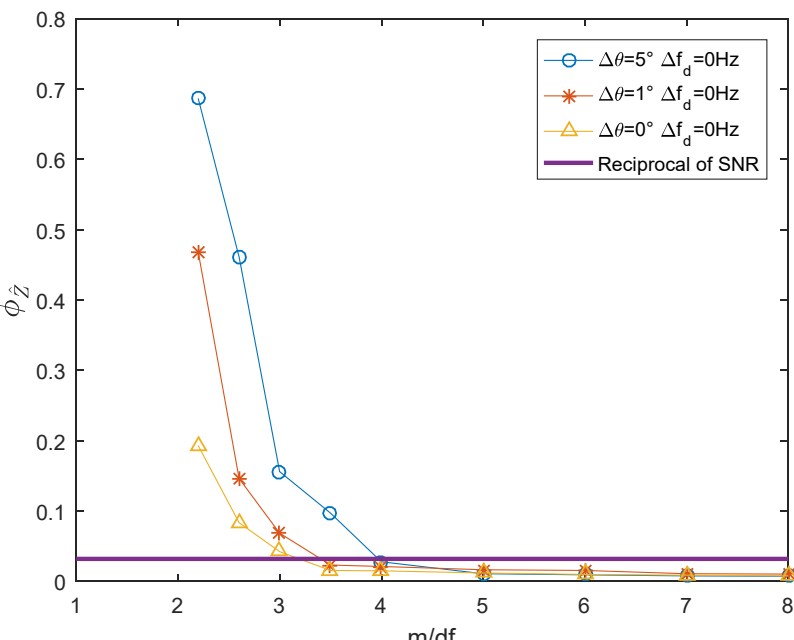

**Figure 4.** Relative matrix recovery errors under different angle separations between two targets while the Doppler frequency separation is set as 0 Hz.

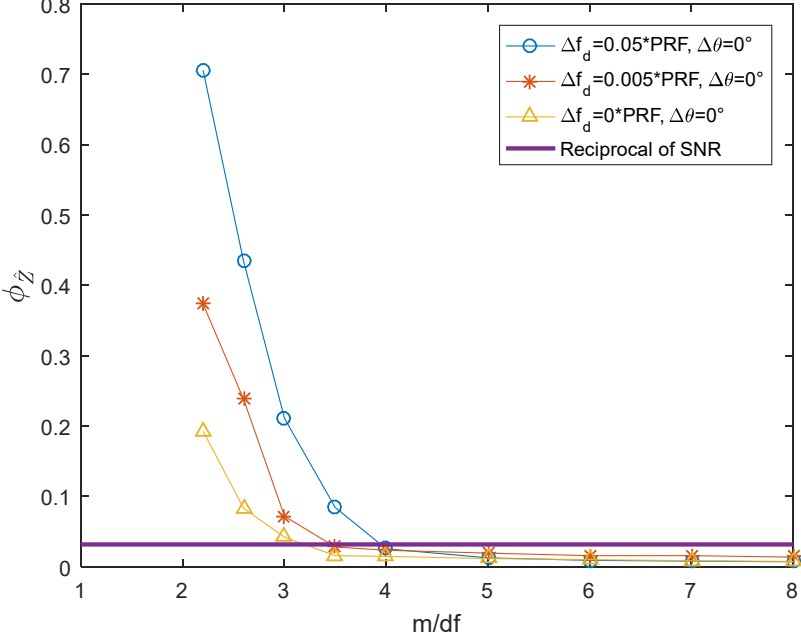

**Figure 5.** Relative matrix recovery errors under different Doppler frequency separations between two targets while the angular separation is set as 0°.

*5.2. Angle-Doppler Frequency Estimation*

Figure 6 shows the sparse target scene on an angle-Doppler frequency map for a 0 dB SNR scenario. This scenario comprises five-point targets, where the leftmost two targets are very close, which locate at $(45°, -0.2)$ and $(45.5°, -0.205)$ in the angle-Doppler frequency domain, respectively. From Figure 6, we can see that the joint angle-Doppler frequencies can be exactly estimated by OMP algorithms based on the recovered matrix.

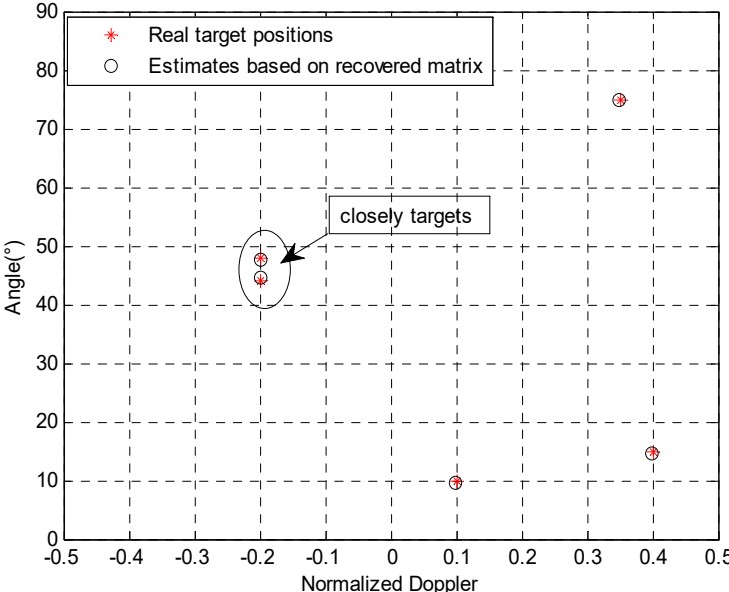

**Figure 6.** Real target positions along with the estimates.

Figure 7 shows the sparse target scene on an angle-Doppler frequency map for a 25 dB SNR scenario. This scenario comprises two point targets, which locate at $(10°, -0.3)$ and $(10°, +0.1)$ in the angle-Doppler frequency domain, respectively. From Figure 7, we can see that the joint angle-Doppler frequencies can be exactly estimated by OMP algorithms based on the recovered matrix.

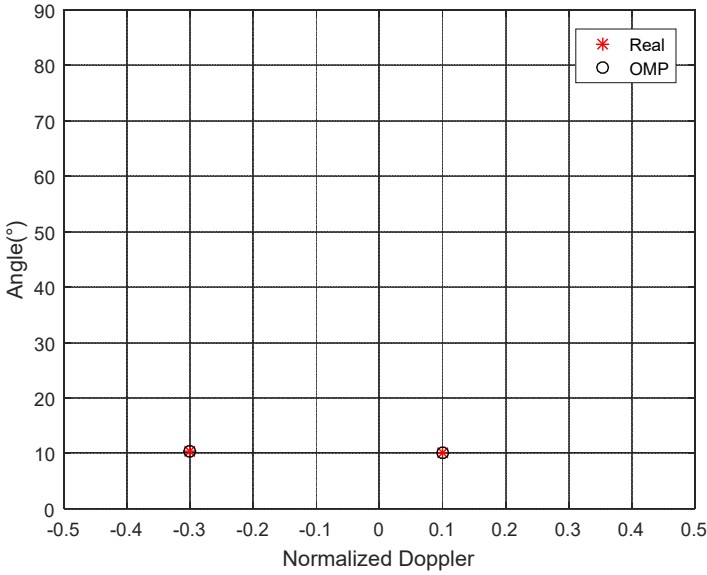

**Figure 7.** Real target positions along with the estimates. OMP: orthogonal matching pursuit.

The performance of high-resolution is provided from both angle and Doppler dimensions. However, for the case where the two targets have identical Doppler frequency but different angles, the estimation performance would decrease. Hence, we examine the angle resolution in terms of the ability to separate two closely spaced targets with equal Doppler frequency. Let us consider the scenario with two targets. The real value and estimated value of the target angle are, respectively, set as $\theta_i$ and $\hat{\theta}_i$, $i = 1, 2$. We set $M = 100$, $N = 100$, $\theta_1 = 30°$, and $\theta_2 = \theta_1 + \Delta\theta$, where $\Delta\theta = [0.5° : 0.5° : 5°]$. If $\|\hat{\theta}_i - \theta_i\| \leq \varepsilon\Delta\theta$, $\varepsilon = 0.1$, we declare the estimation a success, which means the two targets are distinguished successfully from the angle domain; otherwise, they are distinguished unsuccessfully. The

probability of angle resolution is then defined as the fraction of successful events in 200 independents, which is illustrated in Figure 8.

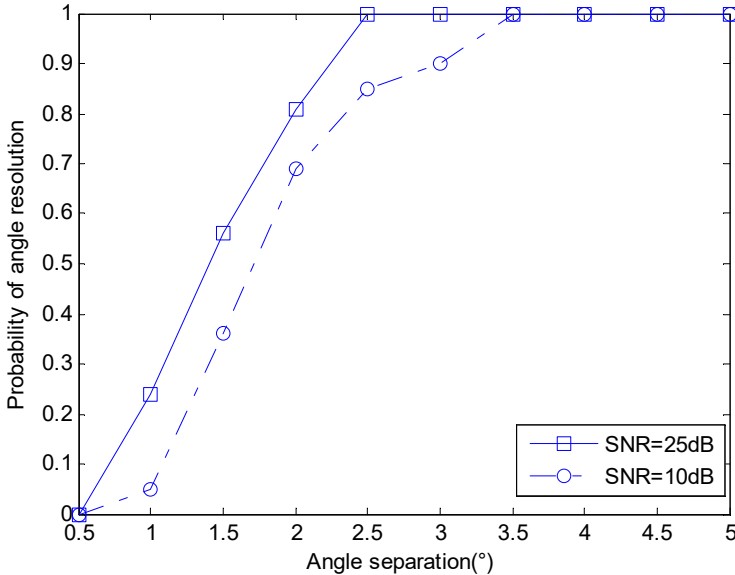

**Figure 8.** The probability of distinguishing two targets with equal Doppler frequency under different angle separations.

## 6. Conclusions

We demonstrate a high-resolution radar approach termed as single-channel sub-Nyquist-MC radar, which employs the sub-Nyquist, matrix completion, as well as compressed sampling techniques. The proposed single-channel sub-Nyquist-MC radar approach allows for minimizing the number of samples in all three dimensions, implying power consumption saving, and hence gaining substantial storage capacity reduction. We compared our proposed radar approach to the conventional MTI radar and have seen its clear advantages in simulations: in a scenario with $K$ point targets in the far field, when the number of samples is reduced from $L \times M \times N$ required by conventional MTI radar to $m$, where $m \approx 4df$ and $df = K(M + N - K)$, the proposed single-channel sub-Nyquist-MC radar is still able to achieve high-resolution angle-Doppler estimation. In future work, we will apply the proposed algorithm to the actual work scene.

**Author Contributions:** Conceptualization, investigation and writing, Q.W.; supervision, review and editing, Y.S.

**Funding:** This work is supported by National Natural Science Funds of China (No. 61703280).

**Conflicts of Interest:** The authors declared that they have no conflict of interest to this work. We declare that we do not have any commercial or associative interest that represents a conflict of interest in connection with the work submitted.

## Appendix A. Matrix Completion

Let us first consider an $n_1 \times n_2$ complex matrix $\mathbf{M}$ of rank $r$, whose singular value decomposition (SVD) is given by $\mathbf{M} = \sum_{i=1}^{r} \rho_i u_i \mathbf{v}_i^H$ and with column and row spaces denoted by $U$ and $V$, which spanned by the set $\left\{ \mathbf{u}_i \in \mathbf{C}^{n_1 \times 1}, i = 1, 2, \cdots, r \right\}$ and $\left\{ \mathbf{v}_i \in \mathbf{C}^{n_2 \times 1}, i = 1, 2, \cdots, r \right\}$, respectively. Then, we can define the coherence of $U$ as

$$\mu(U) = \frac{n_1}{r} \sup_{1 \leq i \leq n_1} \|\mathbf{P}_U \mathbf{e}_i\|_2^2 \in \left[1, \frac{n_1}{r}\right] \tag{A1}$$

where $\mathbf{P}_U$ is the orthogonal projection operator onto $U$ and $\mathbf{e}_i$ denotes the standard basis. Likewise, the coherence of $V$ is defined by $\mu(V)$. Further, two assumptions about matrix $\mathbf{M}$ are introduced in [11] as below

**A0)** The coherence obeys $\max(\mu(U), \mu(V)) \leq \mu_0$ for some positive $\mu_0$.

**A1)** The $n_1 \times n_2$ matrix $\sum_{1 \leq k \leq r} u_k v_k^H$ has a maximum entry bounded by $\mu_1 \sqrt{r/(n_1 n_2)}$ in the absolute value for some positive $\mu_1$.

Moreover, it is notable that **A1)** always holds with $\mu_1 = \mu_0 \sqrt{r}$ if **A0)** holds [12].

Suppose that the matrix $\mathbf{M}$ is corrupted with noise, that is, $\mathbf{Y} = \mathbf{M} + \mathbf{W}$, where $\mathbf{W}$ denotes the noise component. Let us define $\mathcal{P}_{\Omega}(\cdot)$ as an entrywise sampling operator. For example, $\mathcal{P}_{\Omega}(\mathbf{M})$ denotes an entrywise sampling of $\mathbf{M}$, that is, the orthogonal projector onto the span of matrices vanishing outside of $\Omega$ so that the $(i,j)$th component of $\mathcal{P}_{\Omega}(\mathbf{M})$ is equal to $\mathbf{M}_{ij}$ if $(i,j) \in \Omega$ and zero otherwise, where $\Omega$ is the set of indices of observed entries with cardinality $m$. Hence, the observation becomes $\mathcal{P}_{\Omega}(\mathbf{Y}) = \mathcal{P}_{\Omega}(\mathbf{M}) + \mathcal{P}_{\Omega}(\mathbf{W})$. If a matrix $\mathbf{M}$ obeys **A0)** and **A1)**, then $\mathbf{M}$ can be recovered by solving a nuclear norm optimization problem

$$\begin{aligned} &\min \|\mathbf{X}\|_* \\ &\text{s.t. } \|P_{\Omega}(\mathbf{X} - \mathbf{Y})\|_F \leq \delta \end{aligned} \tag{A2}$$

If the noise is zero-mean and white, then parameter $\delta$ can be related to the noise variance $\sigma^2$, that is, $\delta^2 = (m + \sqrt{8m})\sigma^2$. Correspondingly, the recovery error $\|\hat{\mathbf{M}} - \mathbf{M}\|_F$ is bounded as

$$\|\hat{\mathbf{M}} - \mathbf{M}\|_F \leq 4\sqrt{\frac{1}{p}(2+p)\min(n_1, n_2)\delta} + 2\delta \tag{A3}$$

where $p = \frac{m}{n_1 n_2}$ is the fraction of observed entries [14].

The following result gives a probabilistic bound of the number of entries $m$ required to recover the matrix $\mathbf{M}$.

**Theorem 1 [12].** *Suppose that we observe m entries of the rank-r matrix $\mathbf{M} \in \mathbf{C}^{n_1 \times n_2}$ obeying A0) and A1), with matrix coordinates sampled uniformly at random. Let $n = \max\{n_1, n_2\}$. Then, there exist constants C and c such that if*

$$m \geq C \max\left\{\mu_1^2, \mu_0^{1/2}\mu_1, \mu_0 n^{1/4}\right\} nr\beta \log n \tag{A4}$$

*for some $\beta > 2$, then the minimizer to the problem of* (A2) *is unique and equal to $\mathbf{M}$ with probability at least $1 - cn^{-\beta}$. For $r \leq \mu_0^{-1}n^{1/5}$, this estimate can be improved to*

$$m \geq C\mu_0 n^{6/5} r(\beta \log n) \tag{A5}$$

*with the same probability of success. Theorem 1 implies that if the coherence parameter $\mu_0$ is low, few samples are required to recover $\mathbf{M}$.*

## Appendix B. Maximum Coherence of the Spaces

In this appendix, we prove that the maximum coherence of the spaces spanned by the left and right singular vector of $\mathbf{Z}$ in (2). is bounded by the parameter $\mu_0$. From (2), we have $\mathbf{Z} = \mathbf{ADB}^T$. On the one hand, the matrix $\mathbf{Z}$ has the compact singular value decompositions $\mathbf{Z} = \mathbf{U\Sigma V}^H$ with unitaries $\mathbf{U} \in \mathbf{C}^{M \times K}$, $\mathbf{V} \in \mathbf{C}^{N \times K}$, and diagonal matrix $\mathbf{\Sigma} \in \mathbf{R}^{K \times K}$ with nonzero singular values.

On the other hand, let us consider the thin Orthogonal Right (QR) matrix decomposition of $\mathbf{A}$ given by $\mathbf{A} = \mathbf{V}_A \mathbf{G}_A$, where $\mathbf{V}_A \in \mathbf{C}^{M \times K}$ is such that $\mathbf{V}_A^H \mathbf{V}_A \equiv \mathbf{I}_K$ and $\mathbf{G}_A \in \mathbb{C}^{K \times K}$ constitute an upper triangular matrix. Likewise, the QR decomposition of $\mathbf{B}$ is given by $\mathbf{B} = \mathbf{V}_B \mathbf{G}_B$, where $\mathbf{V}_B \in \mathbf{C}^{N \times K}$ is such that $\mathbf{V}_B^H \mathbf{V}_B \equiv \mathbf{I}_K$ and $\mathbf{G}_B \in \mathbf{C}^{K \times K}$ constitute an upper triangular matrix. Then, we have

$\mathbf{Z} = \mathbf{V}_A \mathbf{G}_A \mathbf{D} \mathbf{G}_B^T \mathbf{V}_B^T$, and since the matrix $\mathbf{G}_A \mathbf{D} \mathbf{G}_B^T \in \mathbf{C}^{K \times K}$ is almost surely of full rank by definition, the Singular Value Decomposition (SVD) is given by $\mathbf{G}_A \mathbf{D} \mathbf{G}_B^T = \mathbf{Q}_A \mathbf{\Lambda} \mathbf{Q}_B^H$, where $\mathbf{Q}_A \in \mathbf{C}^{K \times K}$ is such that $\mathbf{Q}_A \mathbf{Q}_A^H = \mathbf{Q}_A^H \mathbf{Q}_A = \mathbf{I}_K$ and $\mathbf{\Lambda} \in \mathbf{R}^{K \times K}$ are non-zero-diagonal, containing the singular values of $\mathbf{G}_A \mathbf{D} \mathbf{G}_B^T$. Thus, the matrix $\mathbf{Z}$ can be rewritten as

$$\mathbf{Z} = \mathbf{V}_A \mathbf{Q}_A \mathbf{\Lambda} \mathbf{Q}_B^H \mathbf{V}_B^T = \mathbf{V}_A \mathbf{Q}_A \mathbf{\Lambda} (\mathbf{V}_B^* \mathbf{Q}_B)^H \tag{A6}$$

which constitutes a valid SVD of $\mathbf{Z}$, since $(\mathbf{V}_A \mathbf{Q}_A)^H \mathbf{V}_A \mathbf{Q}_A \equiv \mathbf{I}_K$.

Consequently, by the uniqueness of the singular values of a matrix, it holds that $\mathbf{\Lambda} \equiv \mathbf{\Sigma}$. Therefore, we can set $\mathbf{U} = \mathbf{V}_A \mathbf{Q}_A$ and $\mathbf{V} = \mathbf{V}_B^* \mathbf{Q}_B$. If $\mathbf{V}_B^n \in \mathbf{C}^{1 \times K}$ and $\mathbf{B}^n \in \mathbf{C}^{1 \times K}$, $n \in (1, N)$ denote the $n$th row of $\mathbf{V}_B$ and $\mathbf{B}$, respectively, it holds that.

$$\mu(V) \leq \frac{N}{\lambda_{\min}(\mathbf{B}^H \mathbf{B})} \tag{A7}$$

Likewise, regarding the coherence of the column space of $\mathbf{Z}$, we get

$$\mu(V) \leq \frac{N}{\lambda_{\min}(\mathbf{B}^H \mathbf{B})} \tag{A8}$$

Due to (A5), a strictly positive lower bound for $\lambda_{\min}(\mathbf{B}^H \mathbf{B})$ is needed to derive, where $\mathbf{B}^H \mathbf{B}$ can be written as

$$\mathbf{B}^H \mathbf{B} \triangleq \begin{bmatrix} N & \delta_{1,0} & \cdots & \delta_{K-1,0} \\ \delta_{1,0}^* & N & \cdots & \delta_{K-1,1} \\ \vdots & \vdots & \ddots & \vdots \\ \delta_{K-1,0}^* & \delta_{K-1,1}^* & \cdots & N \end{bmatrix} \tag{A9}$$

where

$$\delta_{i,j} \triangleq \sum_{n=0}^{N-1} e^{j2\pi T_r n(f_{di} - f_{dj})}, \quad \forall (i,j) \in \mathbf{N}_{K-1} \times \mathbf{N}_{K-1} \tag{A10}$$

Obviously, $N \equiv \delta_{i,i}$, $\forall i \in \mathbf{N}_{K-1}$. Before we proceed with the analysis, let us state the following standard result, which bounds the minimum and maximum eigenvalues of a matrix $\mathbf{M}$ using exclusive functions of the traces of $\mathbf{M}$ and $\mathbf{M}^2$.

**Theorem 3 [12]:** *Let* $\mathbf{M} \in \mathbf{C}^{K \times K}$ *be a matrix with real eigenvalues. Define*

$$\tau \triangleq \frac{\text{tr}(\mathbf{M})}{K} \text{ and } s^2 \triangleq \frac{\text{tr}(\mathbf{M}^2)}{K} - \tau^2 \tag{A11}$$

Then, it is true that

$$\tau - s\sqrt{K-1} \leq \lambda_{\min}(\mathbf{M}) \leq \tau - \frac{s}{\sqrt{K-1}} \tag{A12}$$

$$\tau + \frac{s}{\sqrt{K-1}} \leq \lambda_{\max}(\mathbf{M}) \leq \tau + s\sqrt{K-1} \tag{A13}$$

In order to apply the Theorem, we define $\mathbf{M} \triangleq \mathbf{B}^H \mathbf{B} \in \mathbf{C}^{K \times K}$. The trace of $\mathbf{M}$ is simply $NK$. Hence, we have

$$\tau = \frac{NK}{K} = N \tag{A14}$$

We also need the trace of $\mathbf{M}^2$. Since $\mathbf{M}$ is a Hermitian matrix, it is true that

$$
\begin{aligned}
\mathrm{tr}(\mathbf{M}^2) \quad &= \sum_{k_1=0}^{K-1}\sum_{k_2=0}^{K-1}\left|\delta_{k_1,k_2}\right|^2 \\[2mm]
&\equiv \sum_{k_1=0}^{K-1}\left\{ N^2 + \sum_{\substack{k_2=0\\ k_1\neq k_2}}^{K-1} \left|\sum_{n=0}^{N-1} e^{j2\pi T_r n(f_{dk_1}-f_{dk_2})}\right|^2 \right\} \\[2mm]
&= \sum_{k_1=0}^{K-1}\left\{ N^2 + \sum_{\substack{k_2=0\\ k_1\neq k_2}}^{K-1} \frac{\sin^2\left(\pi N T_r(f_{dk_1}-f_{dk_2})\right)}{\sin^2\left(\pi T_r(f_{dk_1}-f_{dk_2})\right)} \right\} \\[2mm]
&= \sum_{k_1=0}^{K-1}\left\{ N^2 + \sum_{\substack{k_2=0\\ k_1\neq k_2}}^{K-1} \phi_N^2\left[T_r(f_{dk_1}-f_{dk_2})\right] \right\}
\end{aligned}
\tag{A15}
$$

where

$$
\phi_N(x) \triangleq \frac{\sin(\pi N x)}{\sin(\pi x)}, \; x \in \mathbf{R}, \text{ and } N \in \mathbf{N}^+
\tag{A16}
$$

At this point, it is instructive to at least qualitatively study

$$
\mathrm{tr}(\mathbf{M}^2) \le \sum_{k_1=0}^{K-1}\left\{ N^2 + (K-1)\sup_{x\in[\xi_v,0.5]} \phi_N^2(x) \right\} \triangleq KN^2 + K(K-1)\beta_{\xi_v}(N)
\tag{A17}
$$

where

$$
\beta_{\xi_v}(N) = \sup_{x\in[\xi_v,0.5]} \phi_N^2(x),
\tag{A18}
$$

$$
\xi_v \triangleq \min_{\substack{(i,j)\in\mathbb{N}_{K-1}\times\mathbb{N}_{K-1}\\ i\neq j}} g\left(T_r\left|f_{di}-f_{dj}\right|\right)
\tag{A19}
$$

and

$$
g(x) \triangleq \begin{cases} \lceil x\rceil - x, & \lceil x\rceil - x \le \frac{1}{2} \\ x - \lceil x\rceil, & \text{otherwise.} \end{cases}
\tag{A20}
$$

Hence, via Theorem

$$
s^2 \triangleq \frac{\mathrm{tr}(\mathbf{M}^2)}{K} - N^2 \le N^2 + (K-1)\beta_{\xi_v}(N) - N^2 \equiv (K-1)\beta_{\xi_v}(N)
\tag{A21}
$$

Consequently, we can bound $\lambda_{\min}(\mathbf{M}) = \lambda_{\min}(\mathbf{B}^H\mathbf{B})$ from below as

$$
\lambda_{\min}(\mathbf{B}^H\mathbf{B}) \ge N - (K-1)\sqrt{\beta_{\xi_v}(N)}
\tag{A22}
$$

Using (A5) and (A20), we get the upper bound of the coherence of $V$

$$
\mu(V) \le \frac{N}{N - (K-1)\sqrt{\beta_{\xi_v}(N)}}
\tag{A23}
$$

Likewise, regarding the strictly positive lower bound for $\lambda_{\min}(\mathbf{A}^H\mathbf{A})$, we have

$$\lambda_{\min}(\mathbf{A}^H\mathbf{A}) \geq M - (K-1)\sqrt{\beta_{\xi_u}(M)} \tag{A24}$$

where

$$\beta_{\xi_u}(M) = \sup_{x \in [\xi_u, 0.5]} \phi_M^2(x), \tag{A25}$$

$$\xi_u \triangleq \min_{\substack{(i,j) \in \mathbb{N}_{K-1} \times \mathbb{N}_{K-1} \\ i \neq j}} g\left(\frac{d}{\lambda}\left|\sin(\theta_i) - \sin(\theta_j)\right|\right) \tag{A26}$$

The upper bound of the coherence of $U$ becomes

$$\mu(U) \leq \frac{M}{M - (K-1)\sqrt{\beta_{\xi_u}(M)}} \tag{A27}$$

Consequently

$$\max\{\mu(U), \mu(V)\} \leq \max\left\{\frac{M}{M - (K-1)\sqrt{\beta_{\xi_u}(M)}}, \frac{N}{N - (K-1)\sqrt{\beta_{\xi_v}(N)}}\right\} \tag{A28}$$

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
