# Peer review of "A High-Resolution Joint Angle-Doppler Estimation Sub-Nyquist Radar Approach Based on Matrix Completion"

_information, doi:10.3390/info10040124_

Round 1

Reviewer 1 Report

The authors proposed a compressed sensing (or in other words, matrix completion)  based approach to process the "move target indication radar" data, and the angle and Doppler information are going to be extracted. 

However, I have some concerns about their approach:

1. The introduction of of MTI radar in 2.2 is not clear to me. Neither the transmit signal modulation schema nor the signal processing chain is given in a scientific form. Without them, the evaluation of the authors' contribution is then very difficult.

2. It is originally 3D data structure, which contains range, Doppler and angle information. The authors degrade it into 2D problem by reducing the samples along fast time to one. In this case, at certain time point, two targets with similar speeds and angles but different ranges could overlapping with each other. The accuracy of the parameter estimates can be compromised. Would it be a problem to the subsequent warning or tracking algorithm ? 

3. Moreover, the way to reduce the sample along fast time is also not very clear to me, namely what kind of function does the AIC have ? The receive signal is firstly down-converted by the mixer and then integrated over the interval T_r. What is the purpose and the motivation of this processing? After this processing, which property of the signal is changed ? or what features of the signal are prepared in this step, and how would these features be further processed so that the desired signal is reconstructed?  It would be better that the authors present us how the transmit signal looks like. 

4. The reconstruction of sparse data is in practice vulnerable to the noise amount in the receive signal. I am expecting that the authors could present the performance of their algorithm against increasing SNR in the revision. 

5. The authors alleged that this algorithm should work since there are limited number of targets existing in the data. However, the real scenario is very complicated. The assumption could be violated. It is very interesting to see how the performance degrades when the number of targets K increases given a fixed m. Or some similar analysis. I think the authors have tried to do analysis in this direction, e.g. they studied the performance against m/df. However, the physical meaning of m/df is not clearly explained to readers, and df is not a monotone function of K either.

Some detailed comments:

on page 2, line 76, the "CS" is used before defined.

on page 2, line 87, the parameters m and df are just given without any explanation. 

There are lots of blank caskets in the notations. It seems that some special symbols are not recognized by the latex library. This problem must be corrected! It hinders me to understand your algorithm.

P_U should be defined in this paper for ease of understanding . It is also defined in [12]. 

on page 4, line 155, the M and N are interchanged. 

on page 6, line 237, the notation Gpolylog is not defined.

Author Response

Response to Reviewer 1 Comments

Point 1: The introduction of of MTI radar in 2.2 is not clear to me. Neither the transmit signal modulation schema nor the signal processing chain is given in a scientific form. Without them, the evaluation of the authors' contribution is then very difficult..

Response 1:

The introduction of of MTI is described in detail in the literature [1]. Because rigorous mathematical explanations are difficult, this article can only illustrate the effectiveness of the method in experimental form.

Corresponding modification:

No modification

Point 2: It is originally 3D data structure, which contains range, Doppler and angle information. The authors degrade it into 2D problem by reducing the samples along fast time to one. In this case, at certain time point, two targets with similar speeds and angles but different ranges could overlapping with each other. The accuracy of the parameter estimates can be compromised. Would it be a problem to the subsequent warning or tracking algorithm ?

Response 2: 

In this article, Fig .6 shows the sparse target scene on a angle-Doppler frequency map. This scenario comprises five point targets, where the leftmost two targets are very close, we declare the estimation a success, which means the two targets are distinguished successfully from angle domain; otherwise, they are distinguished unsuccessfully.

Corresponding modification:

No modification

Point 3: Moreover, the way to reduce the sample along fast time is also not very clear to me, namely what kind of function does the AIC have ? The receive signal is firstly down-converted by the mixer and then integrated over the interval T_r. What is the purpose and the motivation of this processing? After this processing, which property of the signal is changed ? or what features of the signal are prepared in this step, and how would these features be further processed so that the desired signal is reconstructed?  It would be better that the authors present us how the transmit signal looks like.

Response 3:

In range dimension, we use a single-channel AIC to reduce the number of samples from L to 1; In each receive channel, both the matched filter and high-rate ADC in MTI radars are replaced by an AIC, before which a random switch unit is used to turn on and turn off the AIC. This scheme implies that only one sample can be obtained at each receive node during one pulse when the random switch unit is turned on. According to the proposed sampling scheme, the samples in digital processing center forwarded by all receive nodes and all pulses are only a subset of the entries of a  matrix. If and only if all switch units are turned on, the samples can be arranged into a full matrix of size . When the number of target K is much smaller than the number of receive nodes M and the number of pulses N, the full matrix is low rank. This means that, under certain conditions and with the knowledge of the sampling scheme, the full matrix can be exactly recovered by using matrix completion techniques based on the observations of a small subset of the full matrix. There are several recent papers on matrix completion problem

Corresponding modification:

No modification

Point 4: The reconstruction of sparse data is in practice vulnerable to the noise amount in the receive signal. I am expecting that the authors could present the performance of their algorithm against increasing SNR in the revision.

Response 4: 

In this article, we compared our proposed radar approach to conventional MTI radar and have seen its clear advantages in simulations: in a scenario with K point targets in the far field, when the number of samples is reduced from LxMxN required by conventional MTI radar to m, the proposed single-channel sub-Nyquist-MC radar is still able to achieve high-resolution angle-Doppler estimation.

Corresponding modification:

No modification

Point 5: The authors alleged that this algorithm should work since there are limited number of targets existing in the data. However, the real scenario is very complicated. The assumption could be violated. It is very interesting to see how the performance degrades when the number of targets K increases given a fixed m. Or some similar analysis. I think the authors have tried to do analysis in this direction, e.g. they studied the performance against m/df. However, the physical meaning of m/df is not clearly explained to readers, and df is not a monotone function of K either.

Response 5: 

m/df is the matrix recovery error as function of the number of samples m per degrees of freedom df, a quantity also used in References [19]. The real scenario is very complicated, This article is limited to the length and experimental conditions, it is difficult to do further experiments.

Corresponding modification:

No modification

Point 6: 

on page 2, line 76, the "CS" is used before defined. on page 2, line 87, the parameters m and df are just given without any explanation.

Response 6: 

It has been modified.

Corresponding modification:

At page 2, line 88; At page 2, line 96,

Point 7: 

There are lots of blank caskets in the notations. It seems that some special symbols are not recognized by the latex library. This problem must be corrected! It hinders me to understand your algorithm.

1. P_U should be defined in this paper for ease of understanding . It is also defined in [12].

2. on page 4, line 155, the M and N are interchanged.

3. on page 6, line 237, the notation Gpolylog is not defined.

Response 7: 

It has been modified.

Corresponding modification:

At page 4, line 164; At page 5, line 165; At page 7, line 247,

Reviewer 2 Report

The manuscript presents a sub-Nyquist radar approach based on matrix completion for a high-resolution jointly angle-Doppler estimation. The technique is based on the use of a single-channel analog-to-information convertor (AIC) to reduce the range samples and a bank of random switch units to reduce the spatial-temporal samples.

The novelty of the approach is fair and can meet the interest of the readers.

Author Response

Response to Reviewer 2 Comments

Point 1:

 Page 1 line 2: The title is not effective, the first part and the second part of the title say the same thing. I suggest to change the title by inserting also the high-resolution jointly angle-Doppler estimationconcept.

Response 1:

I think the comments are very good. I have changed the title of the paper to "A high-resolution jointly angle-Doppler estimation sub-Nyquist radar approach based on matrix completion ".

Corresponding modification:

Page 1 line 2.

Point 2: Page 1 line 4: The affiliations line is missed.

Response 2: 

I have added the affiliations line.

Corresponding modification:

Page 1 line 8-11.

Point 3: 

Page 1 line 14:There are too more space tab between byand M, and

andand N.

Response 3:

It has been modified.

Corresponding modification:

At  page 22-24.

Point 4:

 Page 1 line 30: I suggest to replace everywhere LMNwith LxMxN.

Response 4: 

It has been modified.

Corresponding modification:

At page 1-2.

Point 5: 

Page 1 line 40: You have already define this acronym.

Response 5: 

It has been modified.

Corresponding modification:

At Page 2 line 47.

Point 6: Page 1 line 43: You have already define this acronym.

Response 6: 

It has been modified.

Corresponding modification:

At Page 2 line 50.

Point 7: Page 2 line 45: It is not clear the relation between MNand m. I suggest to insert a definition of m.

Response 7: 

It has been modified.

Corresponding modification:

At Page 2 line 52.

Point 8: Page 2 line 71: A space tab is missed between memoryand [19].

Response 8: 

It has been modified.

Corresponding modification:

At Page 2 line 79.

Point 9: Page 2 line 79: A space tab is missed between spaceand Since.

Response 9: 

It has been modified.

Corresponding modification:

At Page 2 line 87.

Point 10: Page 2 line 80: You have already define this acronym.

Response 10: 

It has been modified.

Corresponding modification:

At Page 2 line 88.

Point 11: Page 2 line 79: A space tab is missed between 20and 21.

Response 11: 

It has been modified.

Corresponding modification:

At Page 2 line 88.

Point 12: Page 2 line 96-100: There is not reason to insert these definitions here,

I suggest to delete these line and insert progressively these definitions in text.

Response 12: 

It has been deleted.

Corresponding modification:

At Page 2 line 98-102.

Point 13:Page 3 line 107: I suggest to replace the title Preliminaries with Methods

or something like this.

Response 13: 

It has been modified.

Corresponding modification:

At Page 3 line 115.

Point 14:Page 3 line 113: Two symbols are missed

Response 14: 

Expressed correctly, no need to modify.

Corresponding modification:

Point 15:Page 4 line 126: There are too more space tab between theand (i, j).  Page 4 line 126: Replace (i, j) thwith (i, j)-th.

Response 15: 

It has been modified.

Corresponding modification:

At Page 4 line 134

Point 16:Page 4 line 130-131: After a comma you have to insert a lower case.

Response 16: 

It has been modified.

Corresponding modification:

At Page 4 line 138

Point 17: Page 4 line 134: A space tab is missed between entriesand [14].

Response 17: 

It has been modified.

Corresponding modification:

At Page 4 line 142

Point 18: Page 4 line 135: There are too more space tab between entriesand m.

Response 18: 

Correct format

Corresponding modification:

No modification

Point 19: Page 4 line 137: A symbol is missed.

Response 19: 

It has been modified.

Corresponding modification:

At Page 4 line 146

Point 20: Page 4 line 137-145: I suggest to delete this theorem already presents in

[12] and re-phrase it in text.

Response 20: 

This theorem is an important part of this article, it is difficult to express in text.

Corresponding modification:

No modification

Point 21: Page 4 line 149: You have already define this acronym. Page 4 line 150: You have already define this acronym.

Response 21: 

It has been modified.

Corresponding modification:

At Page 4 line 157-158.

Point 21: Page 4 line 149: You have already define this acronym. Page 4 line 150: You have already define this acronym.

Response 21: 

It has been modified.

Corresponding modification:

At Page 4 line 157-158.

Point 22: Page 5 line 202: You miss a full stop after equation (9). Page 6 line 209: You miss a full stop after equation (10).

Response 22: 

It has been modified.

Corresponding modification:

At Page 6 line 210, 217.

Point 23: Page 6 line 213: I suggest to replace introductionwith Section.

Response 23: 

It has been modified.

Corresponding modification:

At Page 6 line 221.

Point 24: Page 6 line 214-215: What did you mean with in (54) and (47) respectively?

Response 24: 

It has been modified.

Corresponding modification:

At Page 6 line 223.

Point 25: Page 6 line 218: I suggest to replace consequentlywith and consequently.

Response 25: 

It has been modified.

Corresponding modification:

At Page 6 line 223.

Point 26:Page 6 from line 221 and 227 there are some missed symbols.

Response 26: 

I dont think there are some missed symbols.

Corresponding modification:

No modification.

Point 27: Page 6 line 221-228: I suggest to delete this theorem already presents in

[12] and [24] and re-phrase it in text.

Response 27: 

This theorem is an important part of this article, it is difficult to express in text.

Corresponding modification:

No modification.

Point 28: Page 6 line 234: What did you mean with program?

Response 28: 

It has been modified.

Corresponding modification:

At Page 6 line 244.

Point 29: 

Page 6-7: Check Section 3.2 there are many missed symbols.

Response 29: 

It has been modified.

Corresponding modification:

At Page 6 line 252-289.

Point 29: 

Page 7 line 246: Replace Wherewith where

Response 29: 

It has been modified.

Corresponding modification:

At Page 6 line 257.

Point 30: 

Page 8 line 286: You miss a full stop after equation (30).

Response 30: 

It has been modified.

Corresponding modification:

At Page 8 line 297.

Point 31: 

Page 8 line 295-296: Two symbols are missed.

Response 31: 

It has been modified.

Corresponding modification:

At Page 8 line 306-310.

Point 32: 

Page 8 line 307: You have already define this acronym.

Response 32: 

It has been modified.

Corresponding modification:

At Page 9 line 318.

Point 33: 

Page 9 line 321: You miss a full stop after equation (33).

Response 33: 

It has been modified.

Corresponding modification:

At Page 9 line 332.

Point 34: 

Page 9-10 figure 4-figure 5: Replace y-label with ΦZ.

Response 34: 

It has been modified.

Corresponding modification:

At Page 10-11 figure 4-figure 5.

Point 34: 

Page 10 line 327: Check this sentence.

Response 34: 

It has been modified.

Corresponding modification:

At Page 9 line 335, 336.

Point 34: 

Page 11-13: Check Appendix there are many missed symbols.

Response 34: 

It has been modified.

Corresponding modification:

At Page 11-13 Appendix.

Round 2

Reviewer 1 Report

The authors have made a significant modification during the latest revision. I am appreciating that. However, there are still some concerns from my side about this work. I think they should be clarified. 

1. there are still many unrecognized symbols in the notations in this revision, e.g. at the end of line 52 in the brackets after the "m", there is a symbol displayed in blank.

2. In line 90, the authors used the expression "exact estimates". According to my knowledge, the practical application of compressed sensing algorithm can only recover the desired data to a certain degree, namely can only approximate the desired data, since the assumption required in compressed sensing can not always fulfilled and noise is always present. 

3. In line 96, "df = K(M+N-K)" is an empirical formula or a simplified estimation ? Please provide a reference for it.

The proof in section 2.1 is good and necessary. But I think the authors can put most of them in the appendix. 

4. In line 149, I think there is a typo. It should be "minimizer" instead of "minimize".

5. In line 158, it should be "On receiver" instead of "On receive".

6. In section 3, the AIC is introduced. A harmonic signal exp(-j2*pi*k_0*t/T_r) is multiplied with the receive signal. What is the motivation of this operation? The "k_0" is alleged to be an arbitrary integer. How to choose this k_0 ? Is this k_0 identical for all the receive signal ?

In my view, the AIC just averages the receive signal. Am I right ? But I can not get the point of multiplying with  harmonic signal exp(-j2*pi*k_0*t/T_r). 

7. It is given in section 3.1 that "Z" can be decomposed into Doppler and steering components in (7) without giving the signal model. Normally, it should be like that, firstly the transmit signal model is given, and the receive signal can be derived accordingly. The signal after AIC is then also be able to be illustrated in analytical form. To this end, the authors can easily show the readers that the signal Z is able to be decomposed in the form given in (7).  

8. Moreover, only the verification is only done with simulated data. It would be good to show that the proposed algorithm is also able to somehow work in the real scenario.  

Author Response

Response to Reviewer 1 Comments

Point 1:

there are still many unrecognized symbols in the notations in this revision, e.g. at the end of line 52 in the brackets after the "m", there is a symbol displayed in blank.

Response 1:

PDF of the paper has major problems with character-encoding. In many equations and mathematical expressions, some characters appear as blank boxes. The PDF has been regenerated with the new version of the software to solve these problems.

Corresponding modification:

The PDF has been regenerated.

Point 2:

 In line 90, the authors used the expression "exact estimates". According to my knowledge, the practical application of compressed sensing algorithm can only recover the desired data to a certain degree, namely can only approximate the desired data, since the assumption required in compressed sensing can not always fulfilled and noise is always present. 

Response 2: 

The reviewer’s statement is exact, and the articles words are not appropriate.

Corresponding modification:

On page 2, line 90.

Point 3: 

 In line 96, "df = K(M+N-K)" is an empirical formula or a simplified estimation ? Please provide a reference for it. The proof in section 2.1 is good and necessary. But I think the authors can put most of them in the appendix. 

Response 3:

df = K(M+N-K)" is a simplified estimation. References have been added. On page 9, line 329-330 has a specific description. Section 2.1 has been put into the appendix.

Corresponding modification:

Section 2.1 has been put into the appendix. References have been added.

Point 4: 

In line 149, I think there is a typo. It should be "minimizer" instead of "minimize".

Response 4: 

It has been modified.

Corresponding modification:

On page 13, line 417.

Point 5: 

In line 158, it should be "On receiver" instead of "On receive".

Response 5: 

It has been modified.

Corresponding modification:

On page 5, line 154.

Point 6: 

In section 3, the AIC is introduced. A harmonic signal exp(-j2*pi*k_0*t/T_r) is multiplied with the receive signal. What is the motivation of this operation? The "k_0" is alleged to be an arbitrary integer. How to choose this k_0 ?

Is this k_0 identical for all the receive signal ?

In my view, the AIC just averages the receive signal. Am I right ? But I can not get the point of multiplying with harmonic signal exp(-j2*pi*k_0*t/T_r).

Response 6: 

1. k_0 is randomly generated.

2. k_0 is identical for all the receive signal.

3. AIC is a technique for sampling analog signals directly at a rate lower than Nyquist sampling rate. It is a hardware circuit. See:

[1] E. J. Candes and T. Tao. Near-optimal signal recovery from random projections: Universal encoding strategies? IEEE Transactions on Information Theory,52(12):5406–5425, 2007.

[2] Tropp J A . Random Filters for Compressive Sampling[C]// Conference on Information Sciences & Systems. IEEE, 2006.

[3] S. Kirolos, J. Laska, M. Wakin, and M. Duarte. Analog-to-information conversion via random demodulation. In Design, Applications, Integration and Software, 2006 IEEE Dallas/CAS Workshop on, pages 71–74, 2006.

Corresponding modification:

On page 2, line 51-52;  In the reference 9-11.

Point 7: 

It is given in section 3.1 that "Z" can be decomposed into Doppler and steering components in (7) without giving the signal model. Normally, it should be like that, firstly the transmit signal model is given, and the receive signal can be derived accordingly. The signal after AIC is then also be able to be illustrated in analytical form. To this end, the authors can easily show the readers that the signal Z is able to be decomposed in the form given in (7).  

Response 7: 

I think the current expression can clearly explain the problem.

Corresponding modification:

No modification

Point 8:

Moreover, only the verification is only done with simulated data. It would be good to show that the proposed algorithm is also able to somehow work in the real scenario.

Response 8:

At present, our experimental conditions are not enough. In the next step, we will apply the proposed algorithm to the actual work scene.

Corresponding modification:

At page 4, line 164; At page 5, line 165; At page 7, line 247,

Reviewer 2 Report

Please check the appendix, and the text, some missed symbols still remains.

Author Response

Response to Reviewer 2 Comments

Point 1:

Please check the appendix, and the text, some missed symbols still remains.

Response 1:

PDF of the paper has major problems with character-encoding. In many equations and mathematical expressions, some characters appear as blank boxes. The PDF has been regenerated with the new version of the software to solve these problems.

Corresponding modification:

The PDF has been regenerated.

Round 3

Reviewer 1 Report

I am appreciating the quick response of the authors. However, there are still several concerns about the theoretic basis of this work. 

The authors have already put 3 references to support the justification of applying AIC in their work. However, the AIC structure in this work is fundamentally different from the ones in the references. AIC generally is multiplied with a pseudo-random chip sequence for bandwidth spreading. This part is missing in this work. Is there any explanation ?

Another question is whether the arbitrary positive integer k_0 fixed for all the nodes ?

The authors have accepted my suggestion and moved the proofs into appendix. However, this moving is not clean. It leads to the incompletion in presentation, e.g. the sampling operator P is defined in appendix, and directly used in the main text; the notation "E" in line 168 is not defined  It is not allowed since the main text itself should be complete and understandable. 

The performance analysis is not sufficient. This paper is designed to present a high resolution angular method. Thus, the angular performance, i.e. angle estimation error and separation ability over SNR, should be assessed. Moreover, the angle difference in real targets will also influence the performance. This should also be assessed. 

Author Response

Response to Reviewer 1 Comments

Point 1:

The authors have already put 3 references to support the justification of applying AIC in their work. However, the AIC structure in this work is fundamentally different from the ones in the references. AIC generally is multiplied with a pseudo-random chip sequence for bandwidth spreading. This part is missing in this work. Is there any explanation ?

Response 1:

Added the following explanation and a new reference [15].

Random demodulation (RD) [11] and Modulation bandwidth converter (MWC) [15] is the mainstream implementation of AIC, The RD scheme multiplies the signal and the pseudo-random chip sequence by mixing and low-pass filtering. Sampling is then performed by a low speed analog-to-digital converter that is lower than the Nyquist rate. Reference [15] gives a MWC scheme based on compressed sensing. Signals are captured with an analog front end that consists of a bank of multipliers and low-pass filters whose cutoff is much lower than the Nyquist rate.

Corresponding modification:

On page 2, line 57-63.

Point 2:

Another question is whether the arbitrary positive integer k_0 fixed for all the nodes ?

 Response 2: 

This scheme implies that only one sample can be obtained at each receive node during one pulse when the parallel random switch unit is turned on. So integer k_0 is fixed for all the nodes.

Corresponding modification:

On page 2, line 67.

Point 3: 

The authors have accepted my suggestion and moved the proofs into appendix. However, this moving is not clean. It leads to the incompletion in presentation, e.g. the sampling operator P is defined in appendix, and directly used in the main text; the notation "E" in line 168 is not defined  It is not allowed since the main text itself should be complete and understandable.

Response 3:

Modified accordingly as required

Corresponding modification:

On page 5, line 168 and 175.

Point 4: 

The performance analysis is not sufficient. This paper is designed to present a high resolution angular method. Thus, the angular performance, i.e. angle estimation error and separation ability over SNR, should be assessed. Moreover, the angle difference in real targets will also influence the performance. This should also be assessed.

Response 4: 

Added an experiment diagram and a corresponding experiment described .

Corresponding modification:

On page 10, line 303-310.

Reviewer 2 Report

I think that now the paper could be accepted in present form.

Author Response

Thank you for reviewing.

Round 4

Reviewer 1 Report

I have no further comments to the authors. They have revised the manuscript according to my last comments.